# Synthesis and Biological Evaluation of Imidazo[2,1-*b*]Thiazole based Sulfonyl Piperazines as Novel Carbonic Anhydrase II Inhibitors

**DOI:** 10.3390/metabo10040136

**Published:** 2020-03-31

**Authors:** Kesari Lakshmi Manasa, Sravya Pujitha, Aaftaab Sethi, Mohammed Arifuddin, Mallika Alvala, Andrea Angeli, Claudiu T. Supuran

**Affiliations:** 1Department of Medicinal Chemistry, National Institute of Pharmaceutical Education and Research (NIPER), Hyderabad 500 037, India; manasa.kesari@gmail.com (K.L.M.); sravya.pujitha@gmail.com (S.P.); sethiaaftaab@gmail.com (A.S.); arifabib@gmail.com (M.A.); 2Department of Chemistry, Anwarul Uloom College, 11-3-918, New Malleypally, Hyderabad 500001, T. S., India (Present Address); 3Neurofarba Dept., Università degli Studi di Firenze, Sezione di Scienze Farmaceutiche e Nutraceutiche, Via Ugo Schiff 6, 50019 Sesto Fiorentino, Florence, Italy; andrea.angeli@unifi.it

**Keywords:** carbonic anhydrase, hCA I, hCA II, hCA IX, hCA XII, imidazo[2,1-*b*]thiazole, sulfonyl piperazine

## Abstract

A novel series of imidazo[2,1-*b*]thiazole-sulfonyl piperazine conjugates (9aa-ee) has been synthesized and evaluated for carbonic anhydrase (CA, EC 4.2.1.1) inhibitory potency against four isoforms: The cytosolic isozyme hCA I, II and trans-membrane tumor-associated isoform hCA IX and hCA XII, taking acetazolamide (AAZ) as standard drug, using a stopped flow CO_2_ hydrase assay. The results revealed that most of the compounds showed selective activity against hCA II whereas none of them were active against hCA I, IX, XII (K_i_ > 100 µM). The physiologically dominant cytosolic isoform hCA II was inhibited by these molecules with inhibition constants in the range of 57.7–98.2 µM. This new derivative, thus, selectively inhibits hCA II over the hCA I, IX, XII isoforms, which may be used for further understanding the physiological roles of some of these isoforms in various pathologies.

## 1. Introduction

Carbonic anhydrases (CAs, EC 4.2.1.1) are a group of zinc containing metalloenzymes that effectively catalyze the reversible hydration of CO_2_ to bicarbonate and proton, an important reaction for many physiological processes. The human carbonic anhydrases (hCAs) belong to theα- class of carbonic anhydrases having sixteen different form of isoforms. CA isoenzymes play an important role in various cellular processes, such as tumorigenicity, respiration, regulation of pH, electrolytes secretion and many other processes. hCA I and hCA II act as targets for diseases, such as glaucoma and epilepsy, while hCA IX and hCA XII are targets for imaging of tumors and are found to over-express in many cancers. [1,2,3]. Therefore, it is important to develop an effective approach that can selectively inhibit these isoforms involved in different diseases. Nowadays, the major concern of the researchers working in this field is to develop new concept for designing the compounds that are capable of blocking specifically the process catalyzed by one or two isoform ms of these enzymes.

N and S containing fused heterocyclic systems have gained considerable attraction in the field of medicinal chemistry due to their broad spectrum of pharmacological activities [4]. Imidazo[2,1-*b*]thiazole-guanylhydrazone is a promising lead with potent anticancer activity against a number of cancer cell lines [5]. These fused heterocycles exhibit potent anticancer activity through multiple mechanisms [6]. However, a number of imidazo[2,1-*b*][1,3,4]thiadiazole derivatives have been reported to exhibit antitumor activity [7]. Sharma and Supuran et al. have reported a series of benzenesulfonamide bearing imidazothiadiazole and thiazolotriazole hybrids as carbonic anhydrases IX and XII inhibitors [8]. The amide derivatives of probenecid have been reported as selective inhibitors of carbonic anhydrase IX and XII by Carradori et al. [9,10] (Figure 1).

Based on the literature, sulfonamides were the most investigated class of CA inhibitors [11,12]. Acetazolamide (AZA), methazolamide (MZA), ethoxzolamide (EZA), dorzolamide (DZA) (Figure 1) are few of the many sulfonamide-based carbonic anhydrase inhibitors in clinical use [13,14,15]. Pazopanib has been clinically used for several years now whereas indisulam and celecoxib are currently under clinical investigations in various types of tumors (Figure 1) [16]. Supuran et al. have reported various classes of piperazine derivatives as activators of human carbonic anhydrase I, II, IV and VII [17]. Thus, design of novel heterocyclic/aromatic scaffolds with sulfonamide moiety is crucial in developing selective CA inhibitors (CAIs) with potent activity.

Propelled by these findings and continuing our interest in the design of diversified classes of heterocyclic based CAIs, we endeavor to synthesize a new series of imidazo[2,1-*b*]thiazole-sulfonyl piperazine conjugates (Figure 1), which have not been reported earlier. Thus, a novel imidazo[2,1-*b*]thiazole-sulfonyl piperazine hybrids (9aa-ee) were synthesized and demonstrated CA inhibition potential against I, II, IX and XII isoforms of CA using acetazolamide (AAZ) as standard.

## 2. Results and Discussion

### 2.1. Chemistry

The designed imidazo[2,1-*b*]thiazole-sulfonyl piperazine conjugates (9aa-ee) were prepared according to pathways outlined in Scheme 1. Ethyl-2-aminothiazole-4-carboxylate (3) was synthesized by the condensation of ethyl bromopyruvate (2) and thiourea (1) in ethanol under reflux for 4 h. Cyclization of intermediate (3) with various phenacyl bromides (4a-e) in ethanol at reflux temperature resulted in the formation of intermediates (5a–e), which upon ester hydrolysis in the presence of lithium hydroxide monohydrate (LiOH·H_2_O) produced carboxylic acids (6a–e) [18]. Aryl sulfonyl piperazine intermediates (8a-e) were synthesized based on previous reported procedures [19]. Finally, the key intermediate 6a-e with different substitutions were coupled with 8a-e using *N*-(3-dimethylaminopropyl)-*N*′-ethylcarbodiimide hydrochloride (EDCI) and 1-hydroxybenzotriazole (HOBt) in dry dimethylformamide (DMF) conditions to afford the corresponding target compounds in good yields (9aa-ee).

All synthesized compounds (9aa-ee) were characterized by spectral techniques viz. ^1^H NMR, ^13^C NMR and high resolution mass spectrometry (HRMS). The ^1^H NMR spectrum of 9ab displayed characteristic protons of piperazine around *δ*2.8–3.8 ppm. Methyl and Methoxy group of phenyl ring was found to be around *δ* 2.40 ppm and 3.5–4.0 ppm. All the remaining protons appeared in the range of *δ* 6.8–8.20 ppm. ^13^C NMR spectrum of 9ab showed the characteristic carbonyl and methyl group’s at *δ* 158.61, and *δ* 21.02 ppm, respectively and all the remaining carbons appeared in the range of *δ* 108.97–158.56 ppm. The similar pattern was observed for the remaining compounds (9aa-ee).

### 2.2. Carbonic Anhydrase Inhibition

The newly synthesized imidazo[2,1-*b*]thiazole-sulfonyl piperazines (9aa-ee) were screened for their CA inhibition activity against cytosolic (hCA I, hCA II) and the tumor associated (hCA IX, hCAXII) isoforms by a stopped flow CO_2_ hydrase assay [20]. Acetazolamide (AAZ), a clinically used reference standard and the obtained results are represented as K_i_ (µM) are summarized in Table 1. The following structure-activity relationship (SAR) was figured out from the inhibition data of 9aa-ee as shown in Table 1:i.The cytosolic isoforms hCA I and tumor associated isoform hCA IX, hCAXII were not inhibited by the compounds 9aa-ee (K_i_ > 100 µM).ii.The compounds 9aa-ee showed varied inhibitory profiles against cytosolic isoform hCA II. Compound 9ae, 9bb, 9ca, 9cc-ce and 9da elicited inhibitory potencies (K_i_) of 57.7–67.9 µM. Potent compounds (9ca, 9cc-ce) having chlorine (R) at aromatic ring of imidazo thiazole whereas other compounds are ineffective. It is assumed that these compounds, i.e., 9ae (K_i_ = 57.7 µM), 9bb (K_i_ = 76.4 µM), 9ca (K_i_ =79.9 µM), 9cc (K_i_ = 57.8 µM), 9cd (K_i_ = 71.2 µM), 9ce (K_i_ = 62.1 µM), 9da (K_i_ = 67.9 µM) were showing inhibition probably due to the presence of 4-OCH_3_, 4-CH_3_, 4-Cl and 4-F as R group and 4-t-butyl, 4-CH_3_, 4-OCH_3_, 4-Cl and H as R_1_ group.iii.It is evident from the K_i_ values (Table 1), compound 9ae bearing that R-group (imidazothiazole part) substituted with 4-methoxy and R_1_ substituted with 4-t-butyl (sulfonyl piperazine part) showed promising inhibitory activity.iv.Among all the congeners, some of the potent compounds exhibited selective inhibition of hCA II as compared to hCA IX and hCAXII in micromolar range.

## 3. Conclusions

In conclusion, we have reported the synthesis of imidazo[2,1-*b*]thiazole-sulfonyl piperazine conjugates (9aa-ee), which are designed to target CA isoforms. The synthesized compounds (9aa-ee) were screened against the cytosolic isoforms hCA I and II, as well as the transmembrane tumor associated isoforms, hCA IX and XII. Few of the compounds (9ae, 9bb, 9ca, 9cc-ce, 9da and 9dc) exhibited acceptable inhibitory activity against cytosolic isoform hCA II, K_i_ value in the range of 57.7–67.9 µM, while some compounds were found to be weaker inhibitors of tumor associated isoform hCA IX and hCA XII. Hence, further modifications of these newly designed imidazo[2,1-*b*]thiazole-sulfonyl piperazine derivatives (9aa-ee) have the potential to emerge as isoform II selective CAIs. Notably, as for all secondary/tertiary sulfonamides reported to date as CAIs, the inhibition mechanism with these derivatives reported here is for the moment unknown [21,22,23].

## 4. Experimental Section

### 4.1. Materials and Methods

All Chemicals and reagents were purchased from the commercial suppliers Alfa Aesar, Sigma Aldrich and used without further purification. The reaction progress was monitored by Thin layer chromatography (TLC) was performed using pre-coated silica gel 60 F_254_ MERCK. TLC plates were visualized and analyzed by exposure to UV light or iodine vapors and aqueous solution of ninhydrin. Column chromatography was performed with Merck flash silica gel with 60–120 mesh size. Melting points were determined on an Electro thermal melting point apparatus and are uncorrected. Nuclear magnetic resonance spectra for ^1^H NMR were obtained on Avance 300, 400 and 500 MHz and analyzed using Mestrenova software and the chemical shifts are reported in ppm from tetramethylsilane (0 ppm) or the solvent resonance as the internal standard (CDCl_3_ 7.26 ppm, DMSO-*d**_6_* 2.49 ppm) and for ^13^C NMR the chemical shifts are reported in ppm from the solvent resonance as the internal standard (CDCl_3_ 77 ppm, DMSO-*d**_6_* 39.3 ppm). HRMS was performed on a Varian ESI- QTOF instrument Spectral data for all compounds are available in the Appendix A.

#### 4.1.1. General Synthetic Procedure for the Preparation of Compound 3

Compound 3 was synthesized according to the procedures described in the literature [18].

#### 4.1.2. General Synthetic Procedure for the Preparation of Compound 5a-e

Compound 5a-e was synthesized based on previous synthetic procedures [18].

#### 4.1.3. General Synthetic Procedure for the Preparation of Compound 6a-e

Compound 6a-e was synthesized based on previous synthetic procedures [18].

#### 4.1.4. General Synthetic Procedure for the Preparation of Compound 8a-e

Substituted phenyl sulfonyl piperazines 8a-e was synthesized by adding triethylamine (31.5 mmol) slowly to a solution of piperazine in CH_2_Cl_2_ at 0 °C, and then commercially available benzenesulfonyl chloride (7a-e, 11.4 mmol) was added and stirred for 30 min. After the completion of reaction, which was confirmed by TLC (petroleum ether and ethyl acetate (2:1)), the reaction was quenched with water and extracted with CH_2_Cl_2_. The organic layer was dried over anhydrous Na_2_SO_4_ and concentrated on reduced pressure to give 8a-e [19].

#### 4.1.5. General Synthetic Procedure for the Preparation of (5-(aryl)imidazo[2,1-b]thiazol-2-yl)(4-((aryl)sulfonyl)piperazin-1-yl)methanone (9aa-ee)

Above obtained carboxylic acid (6a-e, 1.0 mmol) was dissolved in dry dimethyl formamide (10 mL), cooled to 0 °C (ice bath). EDCI (1.2 mmol), HOBt (1.2 mmol), amine intermediate (8a-e, 1.0 mmol) and triethylamine (3 mmol) were added slowly. After 10 min removed the ice bath and the mixture was stirred at room temperature for 12 h until the starting materials were consumed. Then ice cold water was added to the reaction mixture and extracted with dichloro methane (DCM). The combined organic extracts were washed with aq NaHCO_3_ and dried with Na_2_SO_4_ and concentrated in vacuo. The residue was purified by column chromatography by using ethyl acetate and hexane solvent system to afford the title compounds (9aa-ee).

##### (5-(4-Methoxyphenyl)imidazo[2,1-*b*]thiazol-2-yl)(4-((4-methoxyphenyl)sulfonyl)piperazin-1-yl)methanone (9aa)

White solid: 191mg; 90% yield; mp: 202–204 °C; ^1^H NMR (300 MHz, DMSO-*d_6_*) *δ*: 8.03 (s, 1H), 7.75 (d, J = 8.6 Hz, 2H), 7.68 (d, J = 8.8 Hz, 2H), 7.65 (s, 1H), 7.17 (d, J = 8.8 Hz, 2H), 6.94 (d, J = 8.7 Hz, 2H), 3.85 (s, 3H), 3.77 (s, 7H), 3.00 (s, 4H); ^13^C NMR (126 MHz, DMSO-*d_6_*) *δ*: 162.9, 158.6, 158.5, 147.5, 146.2, 129.8, 126.6, 126.3, 126.0, 124.0, 116.4, 114.6, 114.0, 108.9, 55.7, 55.0, 45.7; MS (ESI): *m/z* 513 [M+H]^+^; HRMS (ESI): *m/z* calcd for C_24_H_25_N_4_O_5_S_2_: 513.12609; found: 513.12467 [M+H]^+^.

##### (5-(4-Methoxyphenyl)imidazo[2,1-*b*]thiazol-2-yl)(4-tosylpiperazin-1-yl)methanone (9ab)

White solid: 186 mg; 88% yield; mp: 216–218 °C; ^1^H NMR (300 MHz, DMSO-*d_6_*) *δ*: 8.03 (s, 1H), 7.76 (d, J = 8.6 Hz, 2H), 7.65 (s, 2H), 7.62 (s, 1H), 7.47 (d, J = 8.0 Hz, 2H), 6.94 (d, J = 8.6 Hz, 2H), 3.77 (s, 7H), 3.01 (s, 4H), 2.41 (s, 3H); ^13^C NMR (101 MHz, DMSO-*d_6_*) *δ*: 158.6, 158.5, 147.5, 146.2, 143.8, 132.0, 129.9, 127.5, 126.6, 126.0, 124.0, 116.4, 114.0, 108.9, 55.0, 45.6, 21.0; MS (ESI): *m/z* 497 [M+H]^+^; HRMS (ESI): *m/z* calcd for C_24_H_25_N_4_O_4_S_2_: 497.13117; found: 497.12979 [M+H]^+^.

##### (1-(4-Methoxyphenyl)-9*H*-pyrido[3,4-*b*]indol-3-yl)(4-tosylpiperazin-1-yl)methanone (9ac)

White solid: 185 mg; 85% yield; mp: 243–245 °C; ^1^H NMR (300 MHz, DMSO-*d_6_*) *δ*: 7.98 (s, 1H), 7.77 (s, 1H), 7.74 (s, 2H), 7.70 (s, 2H), 7.67 (s, 1H), 7.61 (s, 1H), 6.91 (d, J = 8.4 Hz, 2H), 3.77 (s, 7H), 3.07 (s, 4H); ^13^C NMR (75 MHz, CDCl_3_ +DMSO-*d_6_*) *δ*: 158.5, 147.4, 146.2, 138.4, 133.8, 129.4, 129.2, 126.5, 125.9, 123.8, 116.3, 113.8, 108.7, 54.9, 45.4; MS (ESI): *m/z* 517 [M+H]^+^; HRMS (ESI): *m/z* calcd for C_23_H_22_ClN_4_O_4_S_2_: 517.07655; found: 517.07532 [M+H]^+^.

##### (5-(4-Methoxyphenyl)imidazo[2,1-*b*]thiazol-2-yl)(4-(phenylsulfonyl)piperazin-1-yl)methanone (9ad)

White solid: 176 mg; 78% yield; mp: 192–194 °C; ^1^H NMR (300 MHz, DMSO-d_6_) *δ*: 8.03 (s, 1H), 7.76 (d, J = 2.1 Hz, 2H), 7.74 (d, J = 2.2 Hz, 3H), 7.68 (d, J = 7.2 Hz, 2H), 7.65 (s, 1H), 6.94 (d, J = 8.8 Hz, 2H), 3.77 (s, 7H), 3.04 (s, 4H); ^13^C NMR (101 MHz, DMSO-d_6_) *δ*: 159.2, 159.1, 148.1, 146.8, 135.6, 134.0, 130.1, 128.1, 127.2, 126.7, 124.6, 117.0, 114.6, 109.5, 55.7, 46.3; MS (ESI): *m/z* 483 [M+H]^+^; HRMS (ESI): *m/z* calcd for C_23_H_23_N_4_O_4_S_2_: 483.11552; found: 483.11409 [M+H]^+^.

##### (4-((4-(Tert-butyl)phenyl)sulfonyl)piperazin-1-yl)(5-(4-methoxyphenyl)imidazo[2,1-*b*]thiazol-2-yl)methanone (9ae)

White solid: 180 mg; 80% yield; mp: 216–218 °C; ^1^H NMR (300 MHz, CDCl_3_) *δ*: 8.04 (s, 1H), 7.75 (d, J = 8.5 Hz, 2H), 7.67 (d, J = 5.0 Hz, 5H), 6.93 (d, J = 8.6 Hz, 2H), 3.76 (s, 7H), 3.03 (s, 4H), 1.31 (s, 9H); ^13^C NMR (101 MHz, DMSO-d_6_) *δ*: 158.6, 158.5, 156.4, 147.5, 146.2, 132.2, 127.5, 126.6, 126.4, 126.1, 124.0, 116.5, 114.0, 109.0, 55.1, 45.6, 34.9, 30.7; MS (ESI): *m/z* 539 [M+H]^+^; HRMS (ESI): *m/z* calcd for C_27_H_31_N_4_O_4_S_2_: 539.17812; found: 539.17666 [M+H]^+^.

##### (4-((4-Methoxyphenyl)sulfonyl)piperazin-1-yl)(5-(*p*-tolyl)imidazo[2,1-*b*]thiazol-2-yl)methanone (9ba)

White solid: 185 mg; 84% yield; mp: 209–211 °C; ^1^H NMR (500 MHz, DMSO-*d_6_*) *δ*: 8.08 (s, 1H), 7.72 (d, J = 8.1 Hz, 2H), 7.70–7.67 (m, 2H), 7.66 (s, 1H), 7.18 (d, J = 2.8 Hz, 2H), 7.17 (d, J = 3.7 Hz, 2H), 3.85 (s, 3H), 3.76 (s, 4H), 3.00 (s, 4H), 2.30 (s, 3H); ^13^C NMR (101 MHz, DMSO-*d_6_*) *δ*: 162.9, 158.5, 147.6, 146.3, 136.3, 131.1, 129.8, 129.2, 126.3, 124.7, 124.0, 116.7, 114.6, 109.6, 55.7, 45.7, 20.8; MS (ESI): *m/z* 497 [M+H]^+^; HRMS (ESI): *m/z* calcd for C_24_H_25_N_4_O_4_S_2_: 497.13117; found: 497.13004 [M+H]^+^.

##### (5-(*p*-Tolyl)imidazo[2,1-*b*]thiazol-2-yl)(4-tosylpiperazin-1-yl)methanone (9bb)

White solid: 187 mg; 86% yield; mp: 214–216 °C; ^1^H NMR (500 MHz, DMSO-*d_6_*) *δ*: 8.08 (s, 1H), 7.72 (d, J = 8.1 Hz, 2H), 7.66 (s, 1H), 7.64 (d, J = 8.2 Hz, 2H), 7.46 (d, J = 8.1 Hz, 2H), 7.18 (d, J = 8.0 Hz, 2H), 3.76 (s, 4H), 3.01 (s, 4H), 2.41 (s, 3H), 2.30 (s, 3H); ^13^C NMR (101 MHz, DMSO-*d_6_*) *δ*: 158.6, 147.6, 146.3, 143.9, 136.3, 132.0, 131.2, 130.0, 129.2, 127.6, 124.7, 124.0, 116.7, 109.6, 45.6, 21.0, 20.8; MS (ESI): *m/z* 481 [M+H]^+^; HRMS (ESI): *m/z* calcd for C_24_H_25_N_4_O_3_S_2_: 481.13626; found: 481.13497 [M+H]^+^.

##### (4-((4-Chlorophenyl)sulfonyl)piperazin-1-yl)(5-(*p*-tolyl)imidazo[2,1-*b*]thiazol-2-yl)methanone (9bc)

White solid: 178 mg; 80% yield; mp: 195–197 °C; ^1^H NMR (500 MHz, DMSO-*d_6_*) *δ*: 8.09 (s, 1H), 7.76 (d, J = 4.8 Hz, 3H), 7.72 (d, J = 8.0 Hz, 2H), 7.70 (d, J = 1.7 Hz, 1H), 7.65 (s, 1H), 7.18 (d, J = 7.9 Hz, 2H), 3.76 (s, 4H), 3.07 (s, 4H), 2.30 (s, 3H); ^13^C NMR (101 MHz, DMSO-*d_6_*) *δ*: 158.5, 147.6, 146.3, 138.4, 136.3, 133.8, 131.1, 129.7, 129.4, 129.1, 124.7, 124.0, 116.6, 109.5, 45.5, 20.7; MS (ESI): *m/z* 501 [M+H]^+^; HRMS (ESI): *m/z* calcd for C_23_H_22_ClN_4_O_3_S_2_: 501.08164; found: 501.08059 [M+H]^+^.

##### (4-((4-(Tert-butyl)phenyl)sulfonyl)piperazin-1-yl)(5-(*p*-tolyl)imidazo[2,1-*b*]thiazol-2-yl)methanone (9bd)

White solid: 182 mg; 82% yield; mp: 222–224 °C; ^1^H NMR (500 MHz, DMSO-*d_6_*) *δ*: 8.09 (s, 1H), 7.71 (d, J = 8.1 Hz, 2H), 7.68 (s, 4H), 7.67 (s, 1H), 7.17 (d, J = 7.9 Hz, 2H), 3.77 (s, 4H), 3.03 (s, 4H), 2.30 (s, 3H), 1.31 (s, 9H); ^13^C NMR (101 MHz, DMSO-*d_6_*) *δ*: 158.6, 156.4, 147.5, 146.3, 136.3, 132.2, 131.1, 129.1, 127.4, 126.3, 124.7, 124.0, 116.7, 109.6, 45.6, 34.9, 30.7, 20.7; MS (ESI): *m/z* 523 [M+H]^+^; HRMS (ESI): *m/z* calcd for C_27_H_31_N_4_O_3_S_2_: 523.18321; found: 523.18172 [M+H]^+^.

##### (5-(4-Chlorophenyl)imidazo[2,1-*b*]thiazol-2-yl)(4-((4-methoxyphenyl)sulfonyl)piperazin-1-yl)methanone (9ca)

Off White solid: 184 mg; 85% yield; mp: 216–218 °C; ^1^H NMR (300 MHz, DMSO-*d_6_*) *δ*: 8.21 (s, 1H), 7.86 (d, J = 8.5 Hz, 2H), 7.69 (s, 2H), 7.67 (s, 1H), 7.42 (d, J = 8.5 Hz, 2H), 7.17 (d, J = 8.8 Hz, 2H), 3.85 (s, 3H), 3.76 (s, 4H), 3.00 (s, 4H); ^13^C NMR (75 MHz, DMSO-*d_6_*) *δ*: 162.9, 158.4, 144.9, 132.8, 131.4, 129.7, 128.6, 126.4, 124.0, 117.1, 114.6, 110.5, 55.7, 45.6; MS (ESI): *m/z* 517 [M+H]^+^; HRMS (ESI): *m/z* calcd for C_23_H_22_ClN_4_O_4_S_2_: 517.07655; found: 517.07509 [M+H]^+^.

##### (5-(4-Chlorophenyl)imidazo[2,1-*b*]thiazol-2-yl)(4-tosylpiperazin-1-yl)methanone (9cb)

Brown solid: 182 mg; 84% yield; mp: 228–230 °C; ^1^H NMR (500 MHz, DMSO-*d_6_*) *δ*: 8.21 (s, 1H), 7.86 (d, J = 8.6 Hz, 2H), 7.69 (s, 1H), 7.64 (d, J = 8.2 Hz, 2H), 7.47 (d, J = 8.1 Hz, 2H), 7.42 (d, J = 8.6 Hz, 2H), 3.76 (s, 4H), 3.01 (s, 4H), 2.41 (s, 3H); ^13^C NMR (101 MHz, DMSO-*d_6_*) *δ*: 158.5, 148.0, 145.0, 143.9, 132.8, 132.0, 131.5, 130.0, 128.6, 127.6, 126.5, 124.0, 117.2, 110.6, 45.7, 21.0; MS (ESI): *m/z* 501 [M+H]^+^; HRMS (ESI): *m/z* calcd for C_23_H_22_ClN_4_O_3_S_2_: 501.08164; found: 501.08056 [M+H]^+^.

##### (5-(4-Chlorophenyl)imidazo[2,1-*b*]thiazol-2-yl)(4-((4-chlorophenyl)sulfonyl)piperazin-1-yl)methanone (9cc)

Brown solid: 190 mg; 90% yield; mp: 226–228 °C; ^1^H NMR (500 MHz, DMSO-*d_6_*) *δ*: 8.22 (s, 1H), 7.86 (d, J = 6.9 Hz, 2H), 7.76 (s, 3H), 7.69 (d, J = 5.6 Hz, 2H), 7.43 (d, J = 7.0 Hz, 2H), 3.76 (s, 4H), 3.06 (s, 4H); ^13^C NMR (101 MHz, DMSO-*d_6_*) *δ*: 158.5, 148.0, 145.0, 138.5, 133.8, 132.8, 131.5, 129.7, 129.5, 128.6, 126.4, 124.0, 117.1, 110.6, 45.6; MS (ESI): *m/z* 521 [M+H]^+^; HRMS (ESI): *m/z* calcd for C_22_H_19_Cl_2_N_4_O_3_S_2_: 521.02701; found: 521.02595 [M+H]^+^.

##### (5-(4-Chlorophenyl)imidazo[2,1-*b*]thiazol-2-yl)(4-(phenylsulfonyl)piperazin-1-yl)methanone (9cd)

Off White solid: 179 mg; 80% yield; mp: 229–231 °C; ^1^H NMR (300 MHz, DMSO-*d_6_*) *δ*: 8.20 (s, 1H), 7.85 (d, J = 8.1 Hz, 2H), 7.74 (s, 3H), 7.68 (d, J = 6.0 Hz, 3H), 7.42 (d, J = 8.0 Hz, 2H), 3.76 (s, 4H), 3.03 (s, 4H); ^13^C NMR (126 MHz, DMSO-*d_6_*) *δ*: 158.5, 148.0, 145.0, 134.9, 133.5, 132.8, 131.5, 129.6, 128.6, 127.5, 126.4, 124.0, 117.2, 110.6, 45.7, 45.0; MS (ESI): *m/z* 487 [M+H]^+^; HRMS (ESI): *m/z* calcd for C_22_H_20_ClN_4_O_3_S_2_: 487.06599; found: 487.06481[M+H]^+^.

##### (4-((4-(Tert-butyl)phenyl)sulfonyl)piperazin-1-yl)(5-(4-chlorophenyl)imidazo[2,1-*b*]thiazol-2-yl)methanone (9ce)

Cream color solid: 178 mg; 76% yield; mp: 272–274 °C; ^1^H NMR (300 MHz, DMSO-*d_6_*) *δ*: 8.21 (s, 1H), 7.86 (d, J = 8.5 Hz, 2H), 7.70 (s, 1H), 7.68 (s, 4H), 7.42 (d, J = 8.5 Hz, 2H), 3.78 (s, 4H), 3.03 (s, 4H), 1.30 (s, 9H); ^13^C NMR (101 MHz, DMSO-*d_6_*) *δ*: 158.5, 156.4, 148.0, 144.9, 132.8, 132.2, 131.5, 128.6, 127.5, 126.4, 126.4, 124.0, 117.2, 110.7, 45.6, 34.9, 30.7; MS (ESI): *m/z* 543 [M+H]^+^; HRMS (ESI): *m/z* calcd for C_26_H_28_ClN_4_O_3_S_2_: 543.12859; found: 543.12754 [M+H]^+^.

##### (5-(4-Fluorophenyl)imidazo[2,1-*b*]thiazol-2-yl)(4-((4-methoxyphenyl)sulfonyl)piperazin-1-yl)methanone (9da)

Cream color solid: 188 mg; 88% yield; mp: 212–214 °C; ^1^H NMR (300 MHz, DMSO-*d_6_*) *δ*: 8.15 (s, 1H), 7.92–7.82 (m, 2H), 7.73–7.64 (m, 3H), 7.26–7.12 (m, 5H), 3.85 (s, 3H), 3.76 (s, 4H), 3.00 (s, 4H); ^13^C NMR (101 MHz, DMSO-*d_6_*) *δ*: 162.8, 158.4, 147.6, 145.1, 130.3, 130.4, 129.6, 126.5 (d, J = 8.25 Hz), 126.2, 123.9, 116.8, 115.3 (d, J = 21.45 Hz), 114.5, 109.9, 55.6, 45.5; MS (ESI): *m/z* 501 [M+H]^+^; HRMS (ESI): *m/z* calcd for C_23_H_22_FN_4_O_4_S_2_: 501.10610; found: 501.10522 [M+H]^+^.

##### (5-(4-Fluorophenyl)imidazo[2,1-*b*]thiazol-2-yl)(4-tosylpiperazin-1-yl)methanone (9db)

Cream color solid: 185 mg; 84% yield; mp: 238–240 °C; ^1^H NMR (300 MHz, DMSO-*d_6_*) *δ*: 8.15 (s, 1H), 7.87 (dd, J = 8.6, 5.6 Hz, 2H), 7.68 (s, 1H), 7.64 (d, J = 8.2 Hz, 2H), 7.47 (d, J = 8.1 Hz, 2H), 7.20 (t, J = 8.9 Hz, 2H), 3.76 (s, 4H), 3.01 (s, 4H), 2.41 (s, 3H); ^13^C NMR (101 MHz, DMSO-*d_6_*) *δ*: 158.4, 147.6, 145.1, 143.7, 131.9, 130.4(d, J= 2.75 Hz), 129.8, 127.4, 126.6 (d, J = 8.25 Hz), 123.9, 116.8, 115.2, 115.4, 109.9, 45.5, 20.9; MS (ESI): *m/z* 485 [M+H]^+^; HRMS (ESI): *m/z* calcd for C_23_H_22_FN_4_O_3_S_2_: 485.11119; found: 485.11009 [M+H]^+^.

##### (4-((4-Chlorophenyl)sulfonyl)piperazin-1-yl)(5-(4-fluorophenyl)imidazo[2,1-*b*]thiazol-2-yl)methanone (9dc)

Cream color solid: 180 mg; 82% yield; mp: 219–221 °C; ^1^H NMR (300 MHz, DMSO-*d_6_*) *δ*: 8.16 (s, 1H), 7.87 (dd, J = 8.6, 5.6 Hz, 2H), 7.76 (s, 3H), 7.69 (d, J = 7.5 Hz, 2H), 7.20 (t, J = 8.9 Hz, 2H), 3.76 (s, 4H), 3.06 (s, 4H); ^13^C NMR (101 MHz, DMSO-*d_6_*) *δ*: 162.6, 160.2, 158.5, 147.8, 145.3, 138.4, 133.8, 130.5, 129.6 (d, J = 24.5 Hz), 126.7 (d, J = 7.9 Hz), 124.0, 116.9, 115.4 (d, J = 21.5 Hz), 110.0, 45.5, 44.9; MS (ESI): *m/z* 505 [M+H]^+^; HRMS (ESI): *m/z* calcd for C_22_H_19_ClFN_4_O_3_S_2_: 505.05656; found: 505.05542 [M+H]^+^.

##### (5-(4-Fluorophenyl)imidazo[2,1-*b*]thiazol-2-yl)(4-(phenylsulfonyl)piperazin-1-yl)methanone (9dd)

Brown color solid: 175 mg; 76% yield; mp: >350 °C; ^1^H NMR (300 MHz, DMSO-*d_6_*) *δ*: 8.15 (s, 1H), 7.92–7.82 (m, 2H), 7.75 (s, 3H), 7.69 (d, J = 4.3 Hz, 3H), 7.20 (t, J = 8.6 Hz, 2H), 3.77 (s, 4H), 3.04 (s, 4H); ^13^C NMR (101 MHz, DMSO-*d_6_*) *δ*: 162.6, 158.5, 147.8, 145.2, 134.9, 133.4, 130.5, 129.5 (d, J = 6.9 Hz), 127.4 (d, J = 20.0 Hz), 126.6 (d, J = 8.0 Hz), 124.0, 116.9, 115.4 (d, J = 21.5 Hz), 110.0, 45.6, 44.9; MS (ESI): *m/z* 471 [M+H]^+^; HRMS (ESI): *m/z* calcd for C_22_H_20_FN_4_O_3_S_2_: 471.09554; found: 471.09438 [M+H]^+^.

##### (4-((4-(Tert-butyl)phenyl)sulfonyl)piperazin-1-yl)(5-(4-fluorophenyl)imidazo[2,1-*b*]thiazol-2-yl)methanone (9de)

Cream color solid: 175 mg; 78% yield; mp: 244–246 °C; ^1^H NMR (300 MHz, DMSO-*d_6_*) *δ*: 8.16 (s, 1H), 7.87 (dd, J = 8.5, 5.6 Hz, 2H), 7.68 (d, J = 4.6 Hz, 5H), 7.19 (t, J = 8.8 Hz, 2H), 3.78 (s, 4H), 3.03 (s, 4H), 1.31 (s, 9H); ^13^C NMR (101 MHz, DMSO-*d_6_*) *δ*: 162.6, 158.5, 156.4, 147.8, 145.2, 132.2, 130.5, 127.5, 126.6 (d, J = 8.0 Hz), 126.3, 124.0, 117.0, 115.4 (d, J = 21.5 Hz), 110.0, 45.6, 34.9, 30.7; MS (ESI): *m/z* 527 [M+H]^+^; HRMS (ESI): *m/z* calcd for C_26_H_28_FN_4_O_3_S_2_: 527.15814; found: 527.15693 [M+H]^+^.

##### (4-((4-Methoxyphenyl)sulfonyl)piperazin-1-yl)(5-phenylimidazo[2,1-*b*]thiazol-2-yl)methanone (9ea)

Brown color solid: 185 mg; 86% yield; mp: 232–234 °C; ^1^H NMR (300 MHz, DMSO-*d_6_*) *δ*: 8.15 (s, 1H), 7.83 (d, J = 7.5 Hz, 2H), 7.69 (d, J = 6.3 Hz, 3H), 7.37 (t, J = 7.5 Hz, 2H), 7.26 (d, J = 7.2 Hz, 1H), 7.17 (d, J = 8.7 Hz, 2H), 3.85 (s, 3H), 3.76 (s, 4H), 3.00 (s, 4H); ^13^C NMR (101 MHz, DMSO-*d_6_*) *δ*: 162.9, 158.5, 147.8, 146.2, 133.9, 129.8, 128.6, 127.2, 126.3, 124.8, 124.0, 116.9, 114.7, 110.1, 55.7, 45.7; MS (ESI): *m/z* 483 [M+H]^+^; HRMS (ESI): *m/z* calcd for C_23_H_23_N_4_O_4_S_2_: 483.11552; found: 483.11424 [M+H]^+^.

##### (5-Phenylimidazo[2,1-*b*]thiazol-2-yl)(4-tosylpiperazin-1-yl)methanone (9eb)

Cream color solid: 182 mg; 84% yield; mp: 250–252 °C; ^1^H NMR (300 MHz, DMSO-*d_6_*) *δ*: 8.15 (s, 1H), 7.83 (d, J = 7.3 Hz, 2H), 7.68 (s, 1H), 7.64 (d, J = 8.1 Hz, 2H), 7.47 (d, J = 8.1 Hz, 2H), 7.37 (t, J = 7.5 Hz, 2H), 7.25 (t, J = 7.3 Hz, 1H), 3.76 (s, 4H), 3.01 (s, 4H), 2.40 (s, 3H); ^13^C NMR (101 MHz, DMSO-*d_6_*) *δ*: 158.5, 147.8, 146.2, 143.9, 133.9, 132.0, 130.0, 128.6, 127.6, 127.2, 124.7, 124.0, 116.9, 110.1, 45.7, 21.0; MS (ESI): *m/z* 467 [M+H]^+^; HRMS (ESI): *m/z* calcd for C_23_H_23_N_4_O_3_S_2_: 467.12067; found: 467.11942 [M+H]^+^.

##### (4-((4-Chlorophenyl)sulfonyl)piperazin-1-yl)(5-phenylimidazo[2,1-*b*]thiazol-2-yl)methanone (9ec)

Cream color solid: 180 mg; 80% yield; mp: 245–247 °C; ^1^H NMR (300 MHz, CDCl_3_+DMSO-*d_6_*) *δ*: 8.16 (s, 1H), 7.83 (d, J = 7.3 Hz, 2H), 7.76 (d, J = 1.3 Hz, 3H), 7.68 (d, J = 7.7 Hz, 2H), 7.38 (t, J = 7.6 Hz, 2H), 7.25 (t, J = 7.3 Hz, 1H), 3.76 (s, 4H), 3.07 (s, 4H); ^13^C NMR (101 MHz, DMSO-*d_6_*) *δ*: 158.5, 147.8, 146.2, 138.5, 133.9, 133.8, 129.7, 129.5, 128.6, 127.2, 124.7, 124.0, 116.9, 110.1, 45.6, 44.9; MS (ESI): *m/z* 487 [M+H]^+^; HRMS (ESI): *m/z* calcd for C_22_H_20_ClN_4_O_3_S_2_: 487.06599; found: 487.06491 [M+H]^+^.

##### (5-Phenylimidazo[2,1-*b*]thiazol-2-yl)(4-(phenylsulfonyl)piperazin-1-yl)methanone (9ed)

Brown color solid: 175 mg; 75% yield; mp: >350 °C; ^1^H NMR (300 MHz, DMSO-*d_6_*) *δ*: 8.14 (s, 1H), 7.83 (d, J = 7.5 Hz, 2H), 7.76 (d, J = 7.4 Hz, 3H), 7.68 (d, J = 4.7 Hz, 3H), 7.37 (t, J = 7.5 Hz, 2H), 7.26 (d, J = 7.3 Hz, 1H), 3.77 (s, 4H), 3.04 (s, 4H); ^13^C NMR (126 MHz, DMSO-*d_6_*) *δ*: 158.6, 147.8, 146.2, 134.9, 133.9, 133.5, 129.6, 128.6, 127.5, 127.3, 127.1, 124.7, 116.9, 110.1, 45.7, 45.0; MS (ESI): *m/z* 453 [M+H]^+^; HRMS (ESI): *m/z* calcd for C_22_H_21_N_4_O_3_S_2_: 453.10496; found: 453.10394 [M+H]^+^.

##### (4-((4-(Tert-butyl)phenyl)sulfonyl)piperazin-1-yl)(5-phenylimidazo[2,1-*b*]thiazol-2-yl)methanone (9ee)

Cream color solid: 178 mg; 80% yield; mp: 257–259 °C; ^1^H NMR (300 MHz, DMSO-*d_6_*) *δ*: 8.15 (s, 1H), 7.83 (d, J = 7.3 Hz, 2H), 7.69 (s, 1H), 7.68 (s, 4H), 7.37 (t, J = 7.5 Hz, 2H), 7.24 (t, J = 7.3 Hz, 1H), 3.78 (s, 4H), 3.03 (s, 4H), 1.30 (s, 9H); ^13^C NMR (101 MHz, DMSO-*d_6_*) *δ*: 158.6, 156.4, 147.8, 146.1, 133.9, 132.2, 128.6, 127.5, 127.1, 126.4, 124.7, 124.0, 117.0, 110.1, 45.6, 34.9, 30.7; MS (ESI): *m/z* 509 [M+H]^+^; HRMS (ESI): *m/z* calcd for C_26_H_29_N_4_O_3_S_2_: 509.16756; found: 509.16640 [M+H]^+^.

### 4.2. CA Inhibition

The inhibition assay of selected CA isozymes was performed using SX.18V-R Applied Photophysics (Oxford, UK) stopped flow instrument. 10 mM Hepes (pH 7.4) as a buffer, with Phenol Red (at a concentration of 0.2 mM) as an indicator, 0.1M Na_2_SO_4_ or NaClO_4_ (To maintain the constant ionic strength; these anions are not inhibitory in the used concentration), following the CA-catalyzed CO_2_ hydration reaction for a period of 5–10 s. Saturated CO_2_ solutions in water at 25 °C were used as substrate. Stock solutions of inhibitors were prepared at a concentration of 10 mM (in DMSO-water 1:1 *v/v*) and dilutions up to 0.01 nM done with the assay buffer mentioned above. At least 7 different inhibitor concentrations have been used for measuring the inhibition constant. Inhibitor and enzyme solutions were pre-incubated together for 10 min at room temperature prior to assay, in order to allow for the formation of the E-I complex. Triplicate experiments were done for each inhibitor concentration, and the values reported in this paper are the mean of such results. The inhibition constants were obtained by non-linear least square methods using the Cheng-Prusoff equation, as reported earlier, [24,25] and represent the mean from at least three different determinations. All CA isozymes used here were recombinant proteins obtained as reported earlier by our group [26,27,28,29].

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
