# Peer review of "Synthesis and Biological Evaluation of Imidazo[2,1-b]Thiazole based Sulfonyl Piperazines as Novel Carbonic Anhydrase II Inhibitors"

_metabolites, 2020, doi:10.3390/metabo10040136_

Round 1

Reviewer 1 Report

The authors describe the synthesis and evaluation of a panel of imidazo[2,1-b]thiazole derivatives as potential carbonic anhydrase (CA) inhibitors; in particular, cytosolic CA I and II and trans-membrane CA IX and XII are included in the study.

The article is well orientated and written, and according to the spectra in the Supporting Information file, compounds subjected to study herein have high purity. Some of the active compounds show some activity against CA II, which is involved in some interesting diseases, like glaucoma, and also exhibited selectivity towards the rest of the enzymes at the concentration tested. Although the potency is not high (57.7–98.2 μM), I consider it to be a good startpoint for the development of new CA inhibitors, and so the results deserve publication in Metabolites journal upon some minor corrections:

  • Page 3, section 2.2, subsections ii. and iii. Please, modify a little bit the sentecences for better comprehension.
  • Line 127, "tumorassociated"    should be    "tumor associated"
  • Line 129, "will emerges"    should be   "will emerge"
  • Experimental section. I would suggest including more clearly the amount of compound used in section 4.1.5. Is it all the amount obtained in section 4.1.4? So, mmols are recommended to be mentioned in that section. Furtheremore, I would suggest including the mg obtained for each of the compounds described, not only the yield.
  • In section 4.2, it is indicated that "The inhibition constants were obtained by non-linear least square methods usingCheng-Prusoff equation, as reported earlier". Please, provide a reference for that.
  • Reference 2, italics sytle is missing for the volume.
  • Reference 5. There is a "strange" "1" superscript within the reference
  • Reference 6. Spaces are missing before and after the year.
  • In the supporting information, "spectras"  should be "spectra"
  • In the supporting information, MHz and solvent should be indicated for each spectra.

Author Response

Authors are thankful to the editor and reviewers for appreciation, valuable comments and suggestions. The manuscript has now appropriately modified as suggested by the reviewer. The following are the point by point responses to the reservations raised by the reviewer.

Reviewer1:
Point 1: Page 3, section 2, subsections ii. and iii. Please, modify a little bit the sentences for better comprehension.

Response: As suggested by the reviewer, page 3, section 2, subsections 2.1 chemistry part, line no 77 and 81, 82, 84, 89-93  and 2.2 carbonic anhydrase inhibition part, line no 99, 100, 103, 106-108, 111, 115  sentences has been rewrittened .

Point 2: Line 127, "tumorassociated"    should be    "tumor associated".

Response: As suggested by the reviewer, line no 132, tumorassociated" has been modified to "tumor associated".

Point 3: Line 129, "will emerges" should be "will emerge"

Response: As suggested by the reviewer, line no 137, "will emerges" has been modified to "will emerge".

Point 4: Experimental Section. I would suggest including more clearly the amount of compound used in section 4.1.5. Is it the entire amount obtained in section 4.1.4? So, mmols are recommended to be mentioned in that section. Furthermore, I would suggest including the mg obtained for each of the compounds described, not only the yield.

Response: As suggested by the reviewer, in experimental section amount of compound has been included along with yield. In section 4.1.5.  mmols has been included.

Point 5: In section 4.2, it is indicated that "The inhibition constants were obtained by non-linear least square methods using Cheng-Prusoff equation, as reported earlier". Please, provide a reference for that.

Response: As suggested by the reviewer, in section 4.2, CA inhibition part line no 352 reference has been included as 15 and the new reference has been included in the reference list. In line no 353 the numbering of reference has been changed to 17-19.

Point 6: Reference 2, italics sytle is missing for the volume.

Response: As suggested by the reviewer, reference 2, italic style is placed for the volume.

Point 7: Reference 5. There is a "strange" "1" superscript within the reference.

Respon

Point by point response to reviewer’s comments

Authors are thankful to the editor and reviewers for appreciation, valuable comments and suggestions. The manuscript has now appropriately modified as suggested by the reviewer. The following are the point by point responses to the reservations raised by the reviewer.

Reviewer1:
Point 1: Page 3, section 2, subsections ii. and iii. Please, modify a little bit the sentences for better comprehension.

Response: As suggested by the reviewer, page 3, section 2, subsections 2.1 chemistry part, line no 77 and 81, 82, 84, 89-93  and 2.2 carbonic anhydrase inhibition part, line no 99, 100, 103, 106-108, 111, 115  sentences has been rewrittened .

Point 2: Line 127, "tumorassociated"    should be    "tumor associated".

Response: As suggested by the reviewer, line no 132, tumorassociated" has been modified to "tumor associated".

Point 3: Line 129, "will emerges" should be "will emerge"

Response: As suggested by the reviewer, line no 137, "will emerges" has been modified to "will emerge".

Point 4: Experimental Section. I would suggest including more clearly the amount of compound used in section 4.1.5. Is it the entire amount obtained in section 4.1.4? So, mmols are recommended to be mentioned in that section. Furthermore, I would suggest including the mg obtained for each of the compounds described, not only the yield.

Response: As suggested by the reviewer, in experimental section amount of compound has been included along with yield. In section 4.1.5.  mmols has been included.

Point 5: In section 4.2, it is indicated that "The inhibition constants were obtained by non-linear least square methods using Cheng-Prusoff equation, as reported earlier". Please, provide a reference for that.

Response: As suggested by the reviewer, in section 4.2, CA inhibition part line no 352 reference has been included as 15 and the new reference has been included in the reference list. In line no 353 the numbering of reference has been changed to 17-19.

Point 6: Reference 2, italics sytle is missing for the volume.

Response: As suggested by the reviewer, reference 2, italic style is placed for the volume.

Point 7: Reference 5. There is a "strange" "1" superscript within the reference.

Response: As suggested by the reviewer, reference 5, "1" superscript has been removed.

Point 8: Reference 6. Spaces are missing before and after the year.

Response: As suggested by the reviewer, in reference 6, spaces are inserted before and after the year.

Point 9: In the supporting information, "spectras" should be "spectra".

Response: As suggested by the reviewer, in the supporting information "spectras" has been replaced to "spectra".

Point 10: In the supporting information, MHz and solvent should be indicated for each spectra.

Response: As suggested by the reviewer, in the supporting information, MHz and solvent has been included for each spectra.

se: As suggested by the reviewer, reference 5, "1" superscript has been removed.

Point 8: Reference 6. Spaces are missing before and after the year.

Response: As suggested by the reviewer, in reference 6, spaces are inserted before and after the year.

Point 9: In the supporting information, "spectras" should be "spectra".

Response: As suggested by the reviewer, in the supporting information "spectras" has been replaced to "spectra".

Point 10: In the supporting information, MHz and solvent should be indicated for each spectra.

Response: As suggested by the reviewer, in the supporting information, MHz and solvent has been included for each spectra.

Reviewer 2 Report

This manuscript is presenting a series of novel inhibitors of carbonic anhydrases. Inhibitors have been designed based on combination of several functional groups as explained in the first figure. However, the inhibition constants are rather weak. Only several inhibitors exhibited 50-70 micromolar Ki. Such results would be disappointing, but there is some interest because the possible inhibitors do not bear the primary sulfonamide group, a common feature of CA inhibitors. Supplementary materials present NMR spectra for all compounds. It seems that the C spectra have not been attempted to assign. Depite possible ambiguities I would suggest to make this attempt. Please also provide inhibition curves in the supplementary to ensure the presentation of all data.

Some of the phrases and Eglish style should  be corrected:

... will emerges as isoform selective CAIs.

Author Response

Point 1: I would suggest to make this attempt. Please also provide inhibition curves in the supplementary to ensure the presentation of all data.

Response: As per reviewer suggestion, the inhibition graphs are included in supporting information

Point 2:Some of the phrases and Eglish style should  be corrected:

... will emerges as isoform selective CAIs.

Response: The English has been checked andcorrected throughout the manuscript.